

# Integrated metabolome and transcriptome revealed the flavonoid biosynthetic pathway in developing *Vernonia amygdalina* leaves

Lanya Shui[*], Kaisen Huo[*], Yan Chen, Zilin Zhang, Yanfang Li and Jun Niu

Key Laboratory of Genetics and Germplasm Innovation of Tropical Special Forest Trees and Ornamental Plants, College of Forestry, Hainan University, Haikou, Hainan, China

[*] These authors contributed equally to this work.

## ABSTRACT

**Background**. *Vernonia amygdalina* as a tropical horticultural crop has been widely used for medicinal herb, feed, and vegetable. Recently, increasing studies revealed that this species possesses multiple pharmacological properties. Notably, *V. amygdalina* leaves possess an abundance of flavonoids, but the specific profiles of flavonoids and the mechanisms of flavonoid biosynthesis in developing leaves are largely unknown.

**Methods**. The total flavonoids of *V. amygdalina* leaves were detected using ultraviolet spectrophotometer. The temporal flavonoid profiles of *V. amygdalina* leaves were analyzed by LC-MS. The transcriptome analysis of *V. amygdalina* leaves was performed by Illumina sequencing. Functional annotation and differential expression analysis of *V. amygdalina* genes were performed by Blast2GO v2.3.5 and RSEM v1.2.31, respectively. qRT-PCR analysis was used to verify the gene expressions in developing *V. amygdalina* leaves.

**Results**. By LC-MS analysis, three substrates (*p*-coumaric acid, trans-cinnamic acid, and phenylalanine) for flavonoid biosynthesis were identified in *V. amygdalina* leaves. Additionally, 42 flavonoids were identified from *V. amygdalina* leaves, including six dihydroflavones, 14 flavones, eight isoflavones, nine flavonols, two xanthones, one chalcone, one cyanidin, and one dihydroflavonol. Glycosylation and methylation were common at the hydroxy group of C3, C7, and C4' positions. Moreover, dynamic patterns of different flavonoids showed diversity. By Illumina sequencing, the obtained over 200 million valid reads were assembled into 60,422 genes. Blast analysis indicated that 31,872 genes were annotated at least in one of public databases. Greatly increasing molecular resources makes up for the lack of gene information in *V. amygdalina*. By digital expression profiling and qRT-PCR, we specifically characterized some key enzymes, such as Va-PAL1, Va-PAL4, Va-C4H1, Va-4CL3, Va-ACC1, Va-CHS1, Va-CHI, Va-FNSII, and Va-IFS3, involved in flavonoid biosynthesis. Importantly, integrated metabolome and transcriptome data of *V. amygdalina* leaves, we systematically constructed a flavonoid biosynthetic pathway with regards to material supplying, flavonoid scaffold biosynthesis, and flavonoid modifications. Our findings contribute significantly to understand the underlying mechanisms of flavonoid biosynthesis in *V. amygdalina* leaves, and also provide valuable information for potential metabolic engineering.

Corresponding author
Jun Niu, niujun@hainanu.edu.cn

## INTRODUCTION

*Vernonia amygdalina*, belonging to family Asteraceae, is a rapidly regenerating shrub (*Yeap et al., 2010*). This plant is widely distributed in the humid tropical forest of Sub-saharan Africa, Southeast Asia, and southern coastal regions of China (*Igile et al., 1994*). Historically, *V. amygdalina* has been planted as a tropical horticultural crop widely used for medicinal herb, feed, and vegetable in Asia and Africa. Recently, increasing studies revealed that this species possesses multiple pharmacological properties, including antibacterial, antifungal, antiparasite, antimalaria, antihelmintic, antiviral, anticancer, antimutagentic, and antidiabetic activities (*Alara, Abdurahman & Olalere, 2020*; *Asante et al., 2016*; *Atangwho et al., 2013*; *Erasto, Grierson & Afolayan, 2007*; *Yeap et al., 2010*). Thus, phytochemical analysis of *V. amygdalina* has been extensively concerned in the academic field. Notably, sesquiterpenoid lactones, terpenoids, flavonoids, saponins, steroids glycosides, anthraquinone, and coumarins have been characterized in this species (*Ayoola et al., 2008*; *Igile et al., 1994*; *Nwanjo, 2005*; *Tona et al., 2004*).

Flavonoids are classified as plant-specific secondary metabolite, consisting of two phenolic rings linked with one oxane ring (C6-C3-C6). Numerous studies have shown that flavonoids have various biofunctions in plants, such as phototropic response (*Silva-Navas et al., 2016*), protection against ultraviolet-B, cold stress, herbivores, and pathogens (*Dixon & Pasinetti, 2010*). Since the objectively pharmacological activities of flavonoids against many chronic diseases, including neurodegenerative and cancer diseases (*Tohge & Fernie, 2017*), studies of flavonoids have gained popular attention. Interestingly, phytochemical screening of *V. amygdalina* showed abundant flavonoids in leaves, and thus this species is considered as one of natural source of flavonoids (*Alara, Abdurahman & Olalere, 2018*; *Asante et al., 2016*).

It is well known that *p*-coumaroyl-CoA and malonyl-CoA are initial substrates for flavonoid biosynthesis in the cytoplasm. First, as the first rate-limiting enzyme in phenylalanine pathways, phenylalanine ammonia-lyase (PAL) converts phenylalanine into *trans*-cinnamic acid, which is involved in the lignin, lignan, and flavonoid synthesis (*Maeda & Dudareva, 2012*). Then, cinnamic acid 4-hydroxylase (C4H) hydroxylates *trans*-cinnamic acid at the C4 position to produce *p*-coumaric acid (4-coumaric acid) (*Schilmiller et al., 2009*). Last, 4-coumaric acid: CoA ligase (4CL) that is an ATP-dependent enzyme irreversibly catalyzes the *p*-coumaroyl-CoA production (*Hamberger & Hahlbrock, 2004*). In plants, acetyl-CoA is critical material for the biosynthesis of many metabolites. ATP-citrate lyase (ACL) produces the cytoplasmic pool of acetyl-CoA, which can be carbonylated into malonyl-CoA by acetyl-CoA carboxylase (ACC). Compared with plastidial ACC comprising four distinct subunits, cytoplasmic ACC is composed of a large polypeptide with four functional domains, participating in malonyl-CoA formation for flavonoid biosynthesis in cytoplasm (*Yanai et al., 1995*).

When malonyl-CoA and *p*-coumaroyl-CoA have been formed, flavonoid scaffolds can be synthesized by a series of regulatory enzymes (*Saito et al., 2013*). As the first rate-limiting enzyme in the biosynthesis pathway of flavonoids, chalcone synthase (CHS) condenses one molecule of *p*-coumaroyl-CoA and three molecules of malonyl-CoA into naringenin chalcone (*Austin & Noel, 2003*). Subsequently, naringenin chalcone can be stereospecifically cyclized to (2S)-naringenin by chalcone isomerase (CHI) (*Lepiniec et al., 2006*). It is worth noting that the generated (2S)-naringenin can be converted into different flavonoid kinds by different regulatory enzymes. For example, flavone synthase II (FNSII) functions in the first step of the trunk pathway towards flavones (*Ferreyra et al., 2015*). Isoflavone synthase (IFS) catalyzes the isomerization of (2S)-naringenin into the isoflavone biosynthesis (*Jung et al., 2000*). Additionally, flavanone 3-hydroxylase (F3H) catalyzes the oxygenation at C3-postion of (2S)-naringenin to form dihydroflavonols (*Pelletier & Shirley, 1996*). Together, the biosynthesis pathway of flavonoid scaffolds is generally conserved across plant species (*Tohge, De Souza & Fernie, 2017*).

Despite molecular structures of flavonoid scaffolds in different plants are restricted (*Li et al., 2020*; *Liu et al., 2020*), multi-terminal modifications of flavonoids, catalyzed by glycosyltransferase, methyltransferase, phenylacyltransferase, and acyltransferase, result in the chemical diversity of flavonoids (*Saito et al., 2013*). These tailoring modifications affect the biological roles of flavonoids in plants. For example, glycosylation is responsible for the steady accumulation of flavonoids (*Li et al., 2001*) and phenylacetylation enhances UV-B absorbent properties (*Tohge et al., 2016*).

In the current study, to confirm the qualitative and quantitative flavonoid profiles in developing *V. amygdalina* leaves, the leaf samples collected from 10, 20 and 30 days after germination (DAG) were analyzed by LC-MS. Subsequently, the transcriptomic sequencing was perform using Illumina technology. After functional annotation, the genes involved in malonyl-CoA formation, phenylpropanoid pathway, flavonoid biosynthesis, and tailoring reactions of flavonoids were characterized in *V. amygdalina* leaves. Combined with digital expression analysis and qRT-PCR results, the expressed patterns of some key enzymes associated with flavonoid biosynthesis were constructed. Overall, the regulatory enzymes involved in flavonoid biosynthesis were systematically analyzed in developing *V. amygdalina* leaves, which will contribute to the targeted and artificial regulation of specific flavonoid biosynthesis.

## MATERIALS & METHODS

### Plant material

Five plants located in the plantation base of Hainan University (geographical coordinates 110.329463 E; 20.062906 N) were selected. Based on our previous investigation on the developmental process of *V. amygdalina* leaves, the leaves grow rapidly in size before 30 DAG, and then enter the mature stage with slow growth or on change before senescence. Therefore, the leafbuds were marked and the leaves were collected from 10 DAG, 20 DAG and 30 DAG, representing the different developmental phares. Five leaves from each plant were selected each time. The fresh leaves were washed with distilled water and dried, and then were rapidly frozen in liquid nitrogen and stored at −80 °C until use.

## Extraction and determination of total flavonoids

A total of 5 g of fresh *V. amygdalina* leaves were ground into powder and extracted by 70 mL 95% ethanol for 2 h. After filtration, liposoluble substances were removed by solvent extraction with 35 mL petroleum ether. The above 10 mL extracted lipid was added into a 25 mL volumetric flask. The 2.5 mL 30% ethanol and 0.75 mL 5% sodium nitrite solution were added into volumetric flask, mixing and standing for 5 min. Then, 0.75 mL10% aluminum nitrate solution was added, mixing and standing for 5 min. Subsequently, 10 mL 1 mol/L NaOH solution was added and agitated. Last, dilute with 30% ethanol to volume, incubating for 10 min. Three biological repetitions were performed. The 10 mL extracted lipid was added into a 25 mL volumetric flask, and rutin (Solarbio HPLC >98%) was used as standard for manufacturing a standard curve. Ultraviolet spectrophotometer (Persee: TU-1810) was used to set the wavelength at 510 nm to measure the absorbance.

## Metabolite extraction

Leaf samples were ground in liquid nitrogen. The 50 mg sample was immediately transferred in an EP tube with one mL MeOH: CAN: $H_2O$ (4:4:2, V/V) at $-20$ °C precooling. Then, eddy oscillated for 1 min and ultrasonic treated for 40 min at 4 °C. The mixture was incubated at 20 °C 120 min and centrifuged at 4 °C 17000 g for 15 min. The 200 μL supernatant was transferred into a new EP tube, and dried in a vacuum at 35 °C. The sample was dissolved in 400 μL acetonitrile: ddH2O (1:1, v/v), vortex for 60 s and filtrate with 0.22 μm organic membrane. 120 μL supernatant was transferred to a sample bottle for LC-MS analysis. The extraction was replicated three times.

## LC-MS analysis

For metabonomic analysis, the extracted samples were analyzed using Agilent 1290 equipped with ACQUITY UPLC HSS T3 1.8 μm 2.1× 100 mm (Waters). The parameters are described as follows: column temperature, 35 °C; injection volume, 1 μL; solvent A (ddH2O: formic acid, 1000:1, v/v) and solvent B (CH3CN: formic acid, 1000:1, v/v) for positive ion mode; solvent A (2 mM ammonium formate) and solvent B (100% CH3CN) for negative ion mode; flow rate 400 μL/min, gradient: 0.0–0.5 min, 95% solvent A; 0.5–2.0 min, 95–90% solvent A; 2.0–10.0 min, 90–40% solvent A; 10.0 –14.0 min, 40–5% solvent A, 14.0–16.0 min, 5% solvent A, 16.0–18.0 min, 5–95% solvent A, 18.0 –20.0 min, 95% solvent A. MS data were obtained by Agilent 6545QTOF (California, USA). The parameters were as follows: mass scan range 50-1100 m/z; gas temperature 320 °C; gas flow 8 L/min; sheath gas flow 12 L/min; sheath gas temp 320 °C; VCap 3500 V (negative) and 4000 V (positive). MSDIAL3.08 software was used to analyze our data.

## cDNA library sequencing method and process

Total RNA was extracted from samples using RNeasy Plant Mini Kits (Qiagen). mRNA was enriched with Oligo (dT) magnetic beads, and fragmented by adding fragmentation buffer. The fragmented mRNA was used as template to synthesize the first and second cDNA strand with random hexamers. After purification by QiaQuick PCR kit (Qiagen), the poly (A) and sequencing sequence were added. After selection by agarose gel electrophoresis and amplification by PCR, the established cDNA library was sequenced by Illumina

HiSeqTM. To ensure data quality, clean reads were obtained by filtering the original data. All transcriptomic data can be accessed in BioProject PRJNA554198. Trinity v2.2.0 software was used for *de novo* transcriptome assembly. The sequencing was replicated three times for each sample.

## Functional annotation and differential expression analysis

The assembled genes were annotated against the following public databases, TAIR10, Nr (NCBI non-redundant protein sequences), Swiss-Prot and COG (Clusters of Orthologous Groups of proteins) by Blast+ v2.4.0 ( $E$-value $<1e^{-5}$). Gene Ontology (GO) functional classifications and Kyoto Encyclopedia of Genes and Genomes (KEGG) assignments were performed by Blast2GO v2.3.5 (*Götz et al., 2008*) and KOBAS software (*Kanehisa et al., 2008*), respectively. Transcriptome assembly completeness were assessed using the 2,326 conserved genes of eudicots in BUSCO v4.1.2 (Simao, 2015) with the BLAST $E$-value cutoff set to 0.001 (*Simão et al., 2015*).

The expressed levels of genes were estimated using Fragment Per Kilobase of exon model per Million mapped reads (FPKM) by RSEM v1.2.31. Differential expression genes (DEGs) between two groups were performed using edgeR v3.14.0. Genes with false discovery rate (FDR) $<0.01$ and $|\log_2 FC| >1$ were assigned as DEGs.

## qRT-PCR analysis

The above total RNA was used and reversely transcribed using the Reverse Transcription System (Promega). Using PrimerQuest (http://www.idtdna.com/PrimerQuest/Home/Index), the qRT-PCR were designed (Table S1). According to our transcriptome results, 10 housekeeping genes with stable expression levels were selected for stability analysis. Using NormFinder approach, *actin-related protein subunit 3* was regarded as the best stable reference in developing *V. amygdalina* leaves (Fig. S1). The relative expression levels were calculated as $\log_2((1 + E1)^{\Delta Ct1(Control-Sample)}/(1 + E2)^{\Delta Ct2(Control-Sample)})$, E1: PCR efficiency of target-gene primer; E2: PCR efficiency of reference-gene primer; $\Delta Ct1$: the difference of Ct value between control and sample in experimental group; $\Delta Ct2$: the difference of Ct value between control and sample in reference group. The PCR efficiency (E) was estimated from the data obtained from the exponential phase of each individual amplification plot and the equation ($E = 10^{slope}$) (*Ramakers et al., 2003*). The expression levels were analyzed from four biological replicates.

## Statistical analysis

The correlation analysis between the relative contents of flavonoids and gene expressions was preformed using Pearson method. Differences between samples were tested for statistical significance using the Duncan MRT method. Statistical analysis was implemented by SPSS software (version 19.0).

# RESULTS

## The total content of flavonoids in developing *V. amygdalina* leaves

To evaluate flavonoid profiles in developing *V. amygdalina* leaves, the samples were collected at early (10 DAG), middle (20 DAG), and late (30 DAG) development. It was

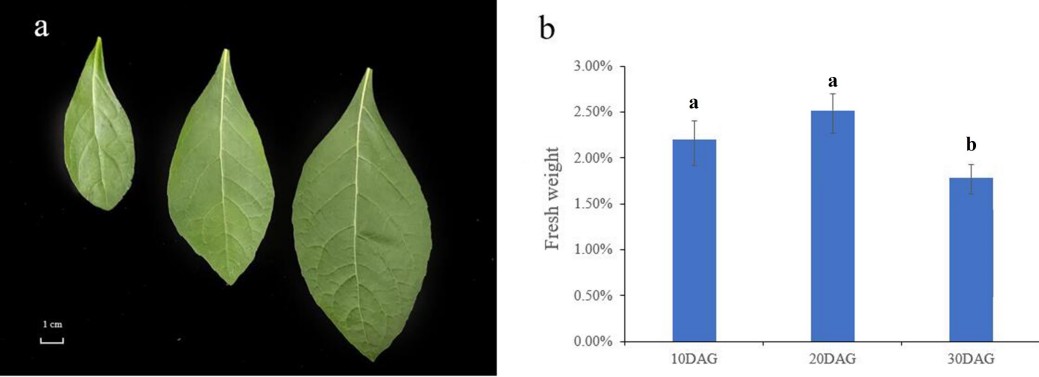

**Figure 1** **The *V. amygdalina* leaves and flavonoid content at different development period.** (A) The *V. amygdalina* leaves at different development period. (B) The total content of flavonoids in developing *V. amygdalina* leaves. Data are reported as mean ± standard deviation. Different letters mean significant difference ($p < 0.01$).

obvious that the leaf size was increased in proportion to the degree of development, for example width ranged from 2.84 ± 0.29 cm to 5.65 ± 0.33 cm, and length ranged from 7.28 ± 0.12 cm to 12.48 ± 0.11 cm (Fig. 1A). Preliminary extraction revealed the presence of flavonoids in developing *V. amygdalina* leaves. The resulting 2.20%, 2.52%, and 1.81% of flavonoid content were identified at 10, 20, and 30 DAG, respectively (Fig. 1B). Compared with mature leaves at 30 DAG, these data indicated a significant accumulation of flavonoids at early-middle (10-20 DAG) development of *V. amygdalina* leaves.

## Flavonoid profiles for *V. amygdalina* leaves

By LC-MS analysis, *p*-coumaric acid, *trans*-cinnamic acid, and phenylalanine involved in transformation pathway of substrate for flavonoid biosynthesis were identified in *V. amygdalina* leaves. The quantitative analysis indicated that phenylalanine and *p*-coumaric acid displayed a similar dynamic pattern, a gradual decline during developing *V. amygdalina* leaves (Table S2 and Fig. 2). *trans*-cinnamic showed a significantly highest relative content at 10 DAG (Table S2 and Fig. 2).

A total of 42 flavonoids were obtained in *V. amygdalina* leaves by LC-MS analysis, including 6 dihydroflavones, 14 flavones, 8 isoflavones, 9 flavonols, 2 xanthones, 1 chalcone, 1 cyanidin, and 1 dihydroflavonol (Table S2). It was observed that diosmetin, nepetin, and luteolin accounted for a relatively large percentage (Table S2), implying that the major flavonoid component in *V. amygdalina* leaves is flavones.

Moreover, the dynamic patterns of different flavonoids showed diversity during leaf development (Fig. 2). For example, nepetin, isorhamnetin, and eriodictyol were preferentially accumulated at 10 DAG, diosmetin, genistein, and prunetin at 20 DAG, and baicalin, tectorigenin, and 7-hydroxyflavonoids at 30 DAG (Table S2). Intriguingly, some flavonoids were specially accumulated at specific period, such as naringenin,

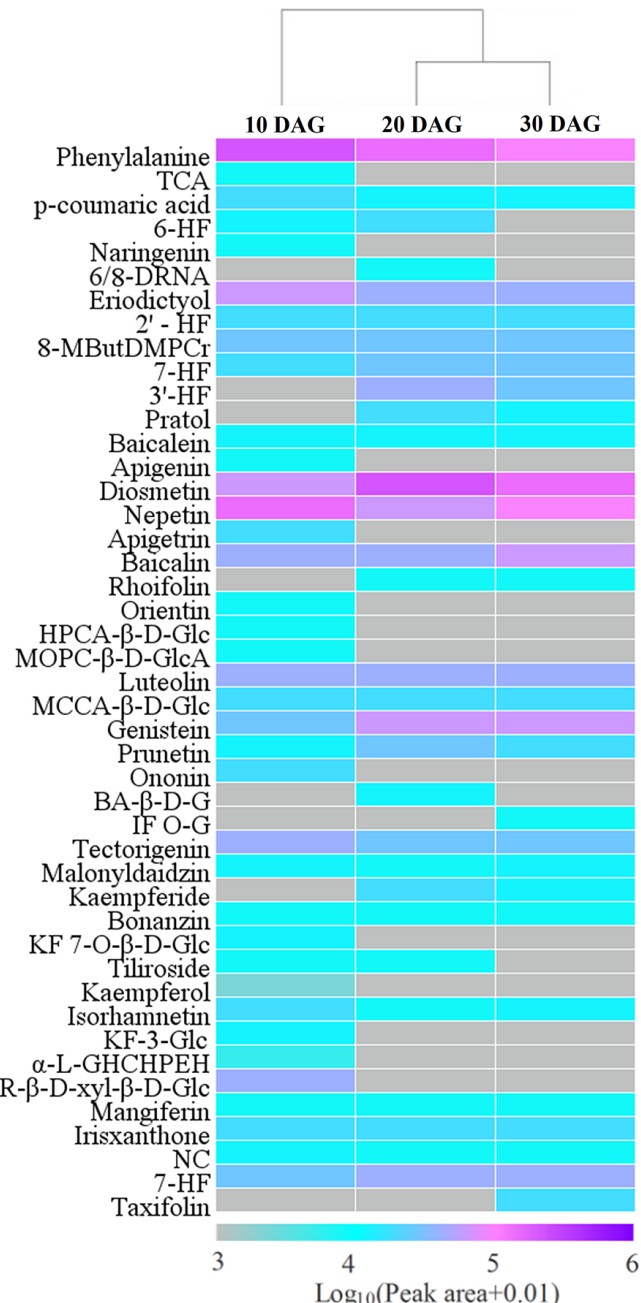

**Figure 2** **Heatmap of flavonoid profiles in *Vernonia amygdalina* leaves.** The data were calculated as $Log_{10}$(Peak area+0.01). TCA, trans-cinnamic acid; 6-HF, 6-Hydroxyflavanone; 6/8-DRNA, 6,8-Diprenylnaringenin; 2′-HF, 2′-Hydroxyflavanone; 8-MButDMPCr, 8-(2,3-Dihydroxy-3-methylbutyl)-5,7-dimethoxy-2-phenyl-2,3-dihydro-4H-chromen-4-one; 7-HF, 7-Hydroxyflavone; 3′-HF, 3′-Hydroxyflavone; HPCA- $\beta$-D-Glc, 5-Hydroxy-2-(4-hydroxyphenyl)-4-oxo-4H-chromen-7-yl 6-O-acetyl- $\beta$-D-glucopyranoside; MOPC- $\beta$-D-GlcA, 5-Hydroxy-2-(3-hydroxy-4-methoxyphenyl)-4-oxo-4H-chromen-7-yl $\beta$-D-glucopyranosiduronic acid; BA- $\beta$-D-G, Biochanin A- $\beta$-D-glucoside; IF O-G, Isoflavonoid O-glycosides; KF 7-O- $\beta$-D-Glc, kaempferol 7-O- $\beta$-D-glucopyranoside; KF-3-Glc, Kaempferol-3-Glucuronide; $\alpha$-L-GHCHPEH, 5-[3-($\alpha$-L-glycero-Hexopyranosyloxy)-5,7-dihydroxy-4-oxo-4H-chromen-2-yl]-2,3-dihydroxyphenyl $\alpha$-L-erythro-hexopyranoside; NC, Naringenin chalcone; 7-HF, 7-hydroxyflavonoids.

apigenin, apigetrin, orientin, ononin, and kaempferol-3-glucuronide at 10 DAG, 6,8-diprenylnaringenin and biochanin A- $\beta$-D-glucoside at 20 DAG, and isoflavonoid O-glycosides and taxifolin at 30 DAG (Fig. 2). These variant favonoid profiles suggest that they may have different functions in *V. amygdalina* leaves.

Importantly, most of flavonoids involved in formation of flavonoid scaffolds were identified in this work, including naringenin chalcone, naringenin, eriodictyol, apigetrin, luteolin, genistein, taxifolin, and kaempferol (Table S2). In addition, it was observed that flavonoid glycosylation and methylation were common in *V. amygdalina* leaves, and generally occurred at the hydroxy group of C3, C7, and C4ʹpositions of flavonoid aglycones, such as Pratol, Diosmetin, Baicalin, Prunetin, kaempferol 7-O-$\beta$-D-glucopyranoside, Biochanin A-$\beta$-D-glucoside, Tiliroside, and Kaempferol-3-Glucuronide (Table S2).

## Transcriptome sequencing and gene assembly

To obtain accurate gene data in developing *V. amygdalina* leaves, transcriptomic libraries were constructed from 10, 20, and 30 DAG and sequenced by illumina platform. More than 5 G data were obtained from each sample, respectively. All clean reads generated from *V. amygdalina* leaves were assembled by the Trinity software, and the resulting 60,422 genes (N50: 1,625 bp) were obtained. From the BLAST results, 26,766 (44.30%), 31,723 (52.50%), and 23,261 (38.50%) genes were functionally annotated in TAIR, Nr, and SwissProt databases, respectively (Table S3). 77.7% of BUSCO's core Eudicots ortholog genes (*Simão et al., 2015*) present in our assembly (Table S4). These data provide abundant genetic resources of *V. amygdalina* for further studies.

## Analysis of differential gene expression

To fully explore the differentially expressed genes in developing *V. amygdalina*, clean reads of each library were mapped into the generated gene database, and the expression levels were calculated by TPM. A total of 2,074 differentially expressed genes were identified (Table S5). The results showed that 10:20 DAG, 10:30 DAG and 20:30 DAG had 1,115 (549 up-regulated and 566 down-regulated genes), 1,619 (804 up-regulated and 815 down-regulated genes) and 467 (169 up-regulated and 298 down-regulated genes) differential genes, respectively (Table S5). By Venn diagram analysis, 79, 246, and 205 were specific DEGs in 10:20, 10:30, and 20:30 DAG, respectively (Fig. 3).

## Formation of malonyl-CoA

ACL and ACC enzymes are responsible for generating the cytosolic pool of acetyl-CoA and malonyl-CoA, respectively. Three *ACL α* (*Va-ACL α1, Va-ACL α2,* and *Va-ACL α3*), one *ACL β* (*Va-ACL β2*), and one ACC (*Va-ACC1*) genes were annotated in *V. amygdalina* (Table S6). Abundant transcript levels of *Va-ACL α1, Va-ACL α3, Va-ACL β2,* and *Va-ACC1* were present in developing *V. amygdalina* leaves, whereas *Va-ACL α2* was expressed at low levels (Fig. 4). Interestingly, the expressions of *Va-ACL α1, Va-ACL α3, Va-ACL β2,* and *Va-ACC1* all showed a coordinated profile with the highest expression at 10 DAG (Fig. 4).

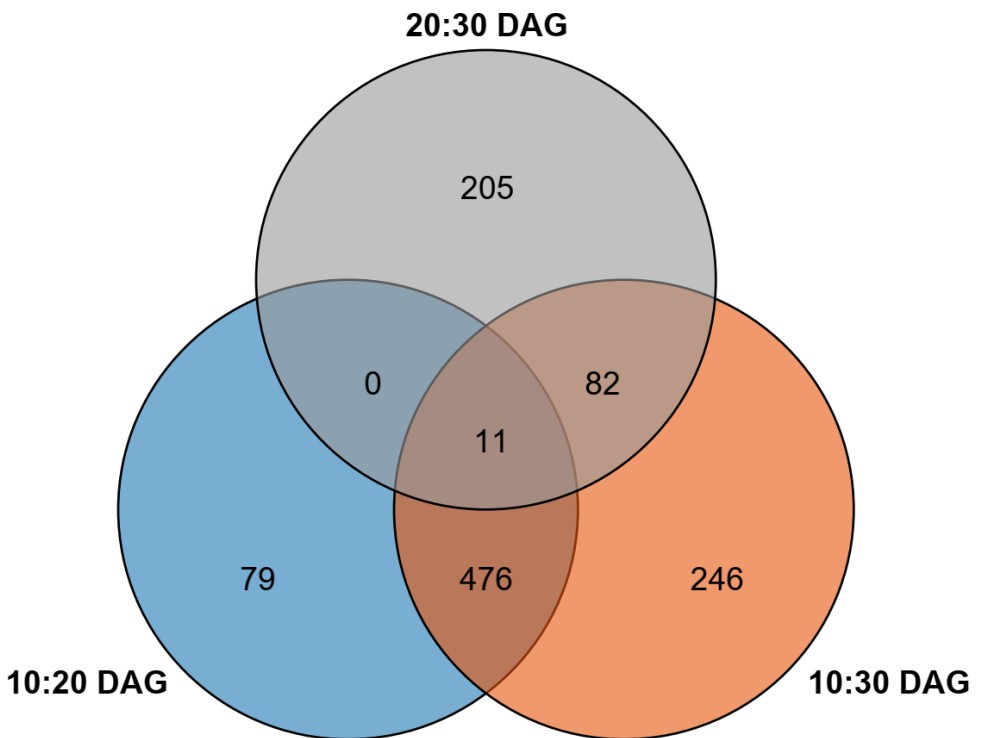

**20:30 DAG**

205

0      82

79      476      246

**10:20 DAG**          **10:30 DAG**

**Figure 3    Number and distribution of differently expressed genes.**

## Formation of *p*-coumaroyl-CoA

PAL is the first committed enzyme in the phenylalanine pathway to provide substrate for flavonoid biosynthesis. In this study, three homologous *PAL* genes were obtained from the *V. amygdalina* transcriptome database (Table S6). By alignment and phylogenetic analysis with *PAL* genes from *Lactuca sativa*, the results showed that these *PAL* genes were conserved (Figs. S2 and S3). The three *Va-PAL* genes all showed highest expressions at 10 DAG, among which the expression levels of *Va-PAL4* and *Va-PAL1* were significantly more abundant than *Va-PAL2* in developing *V. amygdalina* leaves (Figs. 4 and 5).

C4H is a cytochrome P450 monooxygenase, involving in the hydroxylation at C-4 position of *trans*-cinnamic acid. By functional annotation, two homologous *C4H* genes were identified in *V. amygdalina* (Table S6). It is also worth noting that, although both of them had a similar expression pattern in developing *V. amygdalina* leaves, the *Va-C4H1* expression was higher than *Va-C4H2* (Fig. 4).

4CL enzyme catalyzes the CoA-activation of *p*-coumaric acid to form *p*-coumaroyl CoA. Four isoforms of *Va-4CL* genes were identified in *V. amygdalina* (Table S6). Of these, expression levels for *Va-4CL3* were the most abundant isoform in developing *V. amygdalina* leaves (Fig. 4). Moreover, *Va-4CL3* gene exhibited high expressions at 10 and 20 DAG (Figs. 4 and 5).

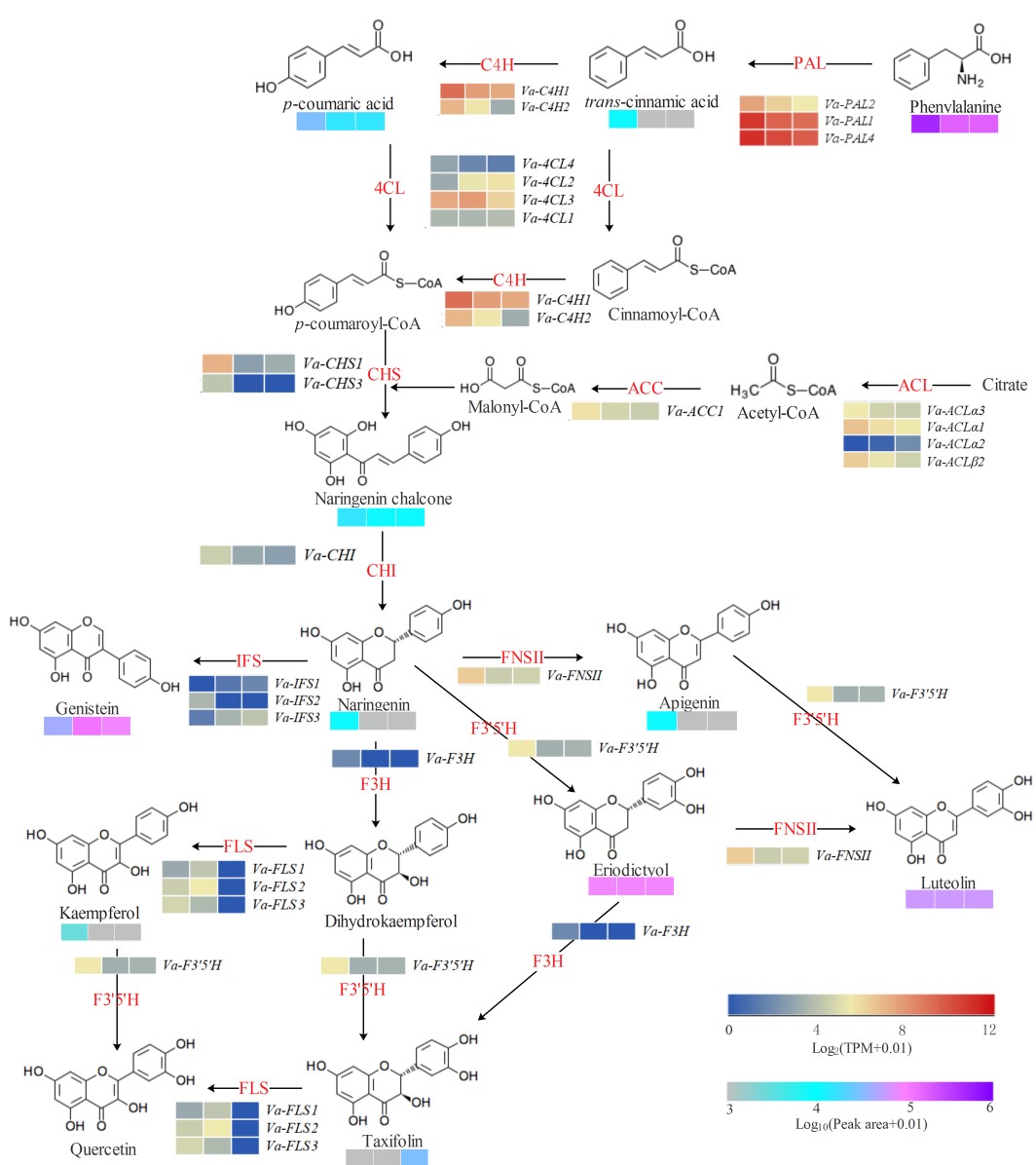

**Figure 4 Biosynthetic pathway of phenylpropanoid and flavonoid skeletons in *Vernonia amygdalina* leaves.** The heatmap from left to right represent the data from 10 DAG, 20 DAG, and 30 DAG, respectively.

## The central flavonoid biosynthetic pathway

CHS is the first committed enzyme in all flavonoid biosynthesis. Two *CHS* genes, *Va-CHS1* and *Va-CHS3* are identified from *V. amygdalina* leaves (Table S6). Although both of them exhibited the highest expression at 10 DAG, *CHS1* gene displayed several-fold higher expression in comparison to *CHS3* gene during the development of *V. amygdalina* leaves (Fig. 4).

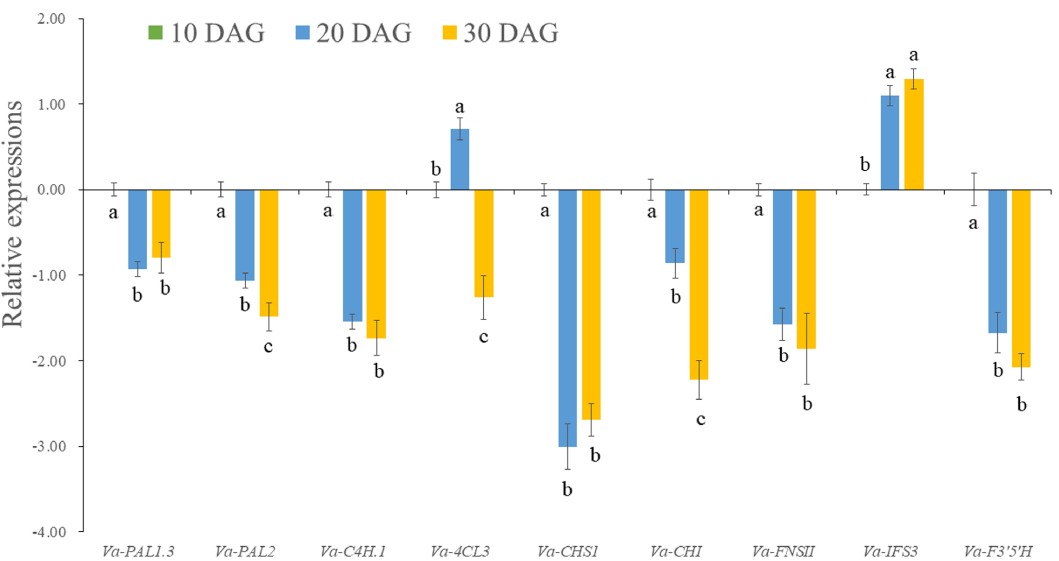

**Figure 5 qRT-PCR analysis of potential genes involved in flavonoid biosynthesis.** The statistical analysis was performed using Duncan method. Data are reported as mean ± standard deviation. Different letters mean significant difference ($p < 0.01$).

CHI catalyzes the stereospecific cyclization of naringenin chalcone to (2S)-naringenin. Only one *Va-CHI* gene was identified from *V. amygdalina* leaves (Table S6). Similar to the expression patterns for *Va-CHS1* genes, *Va-CHI* gene showed the highest expressions at 10 DAG (Figs. 4 and 5).

Interestingly, once flavanones have been formed, they are then converted into different flavonoid kinds by different regulatory enzymes. FNSII catalyzes the conversion of flavanones to flavones, while F3H and IFS are responsible for the conversion of flavanones to isoflavones and dihydroflavonols (Fig. 4), respectively. One *Va-FNSII* and one *Va-F3H* genes were identified in *V. amygdalina* leaves, and displayed an abundant transcript at 10 DAG (Figs. 4 and 5). Three *Va-IFS* genes were found in this work, of which *Va-IFS2* had a high expression at 10 DAG, and the expressions of *Va-IFS3* showed an increase with the leaf development (Fig. 4).

FLS functions in the biosynthesis of flavonols. Three homologous *Va-FLS* genes were obtained (Table S6). During developing *V. amygdalina* leaves, *Va-FLS1* and *Va-FLS2* genes showed up-regulated expressions at 20 DAG, whereas *Va-FLS3* gene displayed the highest expression level at 10 DAG (Fig. 4).

Flavonoid 3′,5′-hydroxylase (F3′5′H) belongs to the cytochrome P-450 family. Although hundreds of genes were annotated as P-450 family (Table S3), only one gene was identified as *Va-F3′5′H* gene involved in 3′- or/and 5′-hydroxylation of flavonoids by KEGG analysis (Table S6 ). Similar to the above genes involved in the central flavonoid biosynthetic pathway, the *Va-F3′5′H* gene also showed the highest expression level at 10 DAG (Figs. 4 and 5).

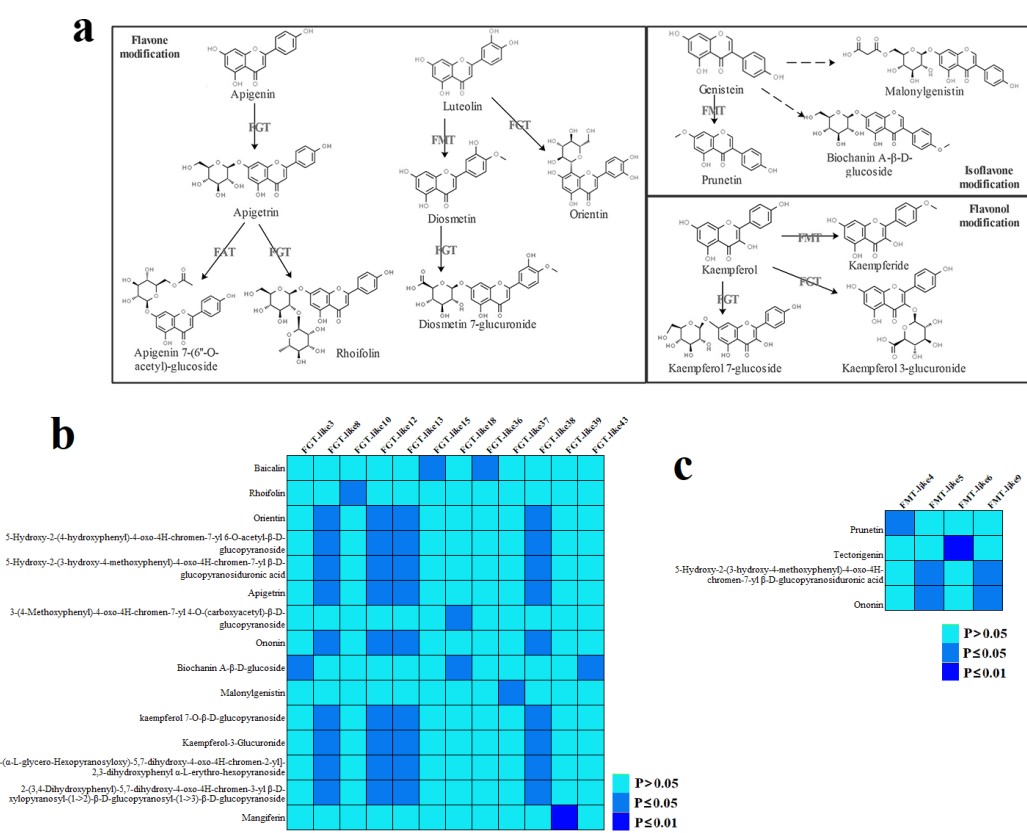

**Figure 6  Flavonoid modifications in *Vernonia amygdalina* leaves.** (A) Proposed pathways of flavonoid modifications. Dotted lines represent this modification pathway needs two steps to complete. (B) These *Va-FGT* genes showed significantly positive correlation with the contents of glycosylated flavonoids in developing *V. amygdalina* leaves. (C) These *Va-FMT* genes showed significantly positive correlation with the contents of methylated flavonoids in developing *V. amygdalina* leaves.

## Tailoring reactions of flavonoid scaffolds

According to the above LC-MS results, the possible modification pathway of flavonoid biosynthesis was constructed (Fig. 6A), reflecting glycosylation and methylation were common modifications in *V. amygdalina* leaves. Based on the homology with known genes of *A. thaliana* involved in glycosylation, methylation, and acetylation of flavonoids, a total of 46, 9, and 18 genes potentially encoding flavonoid glycosyltransferase (FGT), flavonoid methyltransferase (FMT), and flavonoid acyltransferase (FAT) were identified, respectively (Table S7). These gene expressions showed distinctly temporal patterns in developing *V. amygdalina* leaves, suggesting that they may function in different development for the metabolite modifications. Notably, correlation analysis indicated that the expression levels of 12 *Va-FGT* and 4 *Va-FMT* genes showed a positive correlation with the relative contents of some glycosylated and methylated flavonoids during developing *V. amygdalina* leaves, respectively (Figs. 6B, 6C and Table S8). These expression results suggested a possible involvement of the correlated genes in glycosylation or methylation of flavonoids.

## DISCUSSION

Because of various pharmacological properties in *V. amygdalina* leaves, the plant has been widely used in diet and medicine in Africa and Asia (*Yeap et al., 2010*). Previous studies of *V. amygdalina* mainly focused on biological activity and extraction method, indicating that flavonoids are abundant in leaves (*Alara, Abdurahman & Olalere, 2018*; *Ayoola et al., 2008*; *Igile et al., 1994*). However, it is lack of systematical investigation on flavonoid profiles and biosynthetic pathway in developing *V. amygdalina* leaves. During the development of *V. amygdalina* leaves, the flavonoid components were qualitatively and quantificationally analyzed in this study. A total of 42 flavonoids were identified in *V. amygdalina* leaves, mainly including flavones, dihydroflavones, isoflavones, and flavonols (Table S2). These results verify that *V. amygdalina* leaves possess abundant flavonoids, reflecting that *V. amygdalina* is a natural source of flavonoids (*Atangwho et al., 2013*).

A lack of genome information in *V. amygdalina* remains a large obstacle to comprehensively understanding biosynthetic pathway of flavonoids. In an effort to overcome this limitation, we constructed transcriptomic libraries of *V. amygdalina* leaves at 10, 20, and 30 DAG, and sequenced by Illumina HiSeqTM platform. The resulting 60,422 genes were assembled and obtained, and a total of 18,863 reliable unigenes ( $\geq$1000 bp) identified in *V. amygdalina* leaves (Table S3). In addition, 31,872 genes were annotated at least in one of public databases (Table S3) and 2,074 differential expression genes were obtained (Fig. 3). Thus, these molecular resources massively filled and enriched the gene dataset of *V. amygdalina*, which will contribute to advancing the research regarding comparative and functional genomics. Together, the LC-MS and transcriptomic data contribute to further explore on the pathway of flavonoid biosynthesis in *V. amygdalina* leaves (Figs. 4 and 6).

In plants. acetyl-CoA is an important carbon source for the biosynthesis of various metabolites, such as fatty acids, flavonoids, and terpenoids (*Feng et al., 2020*). Previous study indicated that the molecule of CoA moiety is too large to transport across biological membranes, therefore the required acetyl-CoA must be separately synthesized in different subcellular compartments (*Brooks & Stumpf, 1966*). Notably, flavonoid biosynthesis occurs in cytoplasm, and ACL is now known to the ATP-dependent enzyme responsible for generating the cytosolic acetyl-CoA. It was reported that the ACL heteromer is composed of ACL $\alpha$ and ACL $\beta$ (*Fatland et al., 2002*). As expected, three *ACL $\alpha$* (*Va-ACL $\alpha$1*, *Va-ACL $\alpha$2*, and *Va-ACL $\alpha$3*) and one *ACL $\beta$* (*Va-ACL $\beta$2*) genes was identified in *V. amygdalina* leaves (Table S6). Expression analysis in developing *V. amygdalina* leaves indicated a closely coordinated expression between *Va-ACL $\alpha$1/3* and *Va-ACL $\beta$2* (Fig. 4). Thus, it is tempting to speculate that the combination of *Va-ACL $\alpha$1/3* and *Va-ACL $\beta$2* may be critical for ACL activity in *V. amygdalina* leaves, as in previous study that only *ACL $\alpha$* or *ACL $\beta$* transcript would lead to a defect in ACL function (*Fatland & Wurtele, 2005*).

Once the acetyl-CoA has been produced in cytoplasm, ACC can catalyze the carboxylation of acetyl-CoA into malonyl-CoA that is shared between flavonoid biosynthesis and the elongation of very long chain fatty acids (C $\geq$20) (*Alban, Job &*
*Douce, 2000*). Previous studies have confirmed that *A. thaliana* has three ACC isozymes, of which a heteromeric form is composed of four distinct subunits (At5g16390, At5g35360, At2g38040, and AtCg00500) located in plastids, while other two heteromeric forms of ACC1 (At1g36160) and ACC2 (At1g36180) comprise a large polypeptide located in cytoplasm and chloroplast, respectively (*Li et al., 2011*). Thus, ACC1 is responsible for the generation of cytosolic malonyl-CoA for flavonoid biosynthesis in *A. thaliana* (*Saito et al., 2013*). Indeed, one obvious homolog (*Va-ACC1*) of *At-ACC1* was found in *V. amygdalina* leaves (Table S6), and *Va-ACC1* gene was co-expressed with *Va-ACL α1/3* and *Va-ACL β2* genes (Fig. 4). Thus, *Va-ACC1* may control the carbon flux into malonyl-CoA for flavonoid biosynthesis in *V. amygdalina* leaves.

In addition to three molecules of malonyl-CoA, the flavonoid biosynthesis also needs one molecule of *p*-coumaroyl CoA. The formation of *p*-coumaroyl CoA must be orderly catalyzed by PAL, C4H, and 4CL enzymes (*Ferreyra et al., 2012*). Although there were three PAL isoforms, the expression levels of *Va-PAL1* and *Va-PAL4* were the most abundant isoforms in developing *V. amygdalina* leaves (Fig. 4). Thus, they may partially complement the function of each other in conversion of phenylalanine to *trans*-cinnamic, as was reported in *A. thaliana* (*Ohl et al., 1990*). Two orthologs (*Va-C4H1* and *Va-C4H2*) of *A. thaliana* C4H (At2g30490) were characterized in this study, but the expressions of *Va-C4H1* were 5 times higher than that of *Va-C4H2* (Table S6). Our data suggest that Va-C4H1 is presumed to represent the principal isoform responsible for the formation of *p*-coumaric acid in *V. amygdalina* leaves. Among the four 4CL isoforms, only the *Va-4CL3* gene was homologous with At-4CL3 (At1g65060), and showed the highest expression than the other three *Va-4CL* genes (Table S6). These data imply that Va-4CL3 may be the major enzyme catalyzing *p*-coumaroyl-CoA production, in agreement with the previous view in *A. thaliana* (*Ehlting et al., 1999*). Together, the expression levels of the above interested genes showed a pattern that correlated with the biosynthesis of their corresponding production (Figs. 4 and 5). Thus, they may collaborate to facilitate the *p*-coumaroyl-CoA production destined to flavonoid biosynthesis in *V. amygdalina* leaves.

As a type III polyketide synthase enzyme, CHS catalyzes the Claisen-ester condensation that is the first committed rate-limiting step in the flavonoid biosynthesis (*Austin & Noel, 2003*). Here, two CHS isoforms of *Va-CHS1* and *Va-CHS3* were the orthologs with *A. thaliana* At-CHS (At5g13930), which has been verified to be involved in first Claisen-ester condensation of flavonoid biosynthesis (*Shirley et al., 1995*). The expression levels of *Va-CHS1* gene were more abundant than *Va-CHS3* gene in developing *V. amygdalina* leaves, suggesting a crucial role of *Va-CHS1* in the synthesis of naringenin chalcone. Additionally, the homology of *At-CHI* (At3g55120) that participates in stereospecific cyclization for (2S)-naringenin formation (*Saito et al., 2013*) was found in *V. amygdalina* leaves as *Va-CHI*. Interestingly, the expressions of *Va-CHI* gene were comparable to that of *Va-CHS1* gene (Fig. 5) and displayed a coordinated temporal pattern with the naringenin accumulation in developing *V. amygdalina* leaves (Fig. 4). Thus, Va-CHI enzyme is a possible candidate related to (2S)-naringenin biosynthesis in *V. amygdalina* leaves.

It is notable that the generated flavanones would have different fates catalyzed by different regulatory enzymes (Fig. 4). One of pathway is flavone biosynthesis catalyzed

by FNS, which dehydrogenize flavanones into flavones at C2 and C3 positions. In plants, there are two distinct enzyme systems of FNS I and FNS II , of which FNS I is mainly distributed in the Apiaceae family (*Martens et al., 2001*), while FNS II widely distributed in many families of higher plants (*Wu et al., 2016*). Sure enough, only one *Va-FNSII* gene was identified in *V. amygdalina* leaves, and its expression exhibit a similar pattern with apigenin contents in developing *V. amygdalina* leaves (Fig. 4). Thus, Va-FNS II may be the key enzyme responsible for flavone formation in *V. amygdalina* leaves. In addition, there is another pathway of isoflavone biosynthesis, in which IFS catalyzes the transference of aromatic group from 2-position to 3-position (*Misra et al., 2010*). Three *Va-IFS* isoforms were found in *V. amygdalina* leaves (Table S6). Notably, the expression levels of *Va-IFS3* showed a pattern that correlated with genistein biosynthesis (Fig. 4), implying an important role of Va-IFS3 in this metabolic process.

F3H and FLS catalyze the last branch of the trunk pathway towards flavonol biosynthesis (Fig. 4). Although the homology of *At-F3H* (At3g51240) was found in *V. amygdalina* leaves, its expression was either expressed at low levels or absent (Fig. 4). Thus, it is tempting to speculate that Va-IFS3 may be not essential for the oxygenation at 3-postion of flavanone. Previous studies in *A. thaliana* decided that the activity of F3H *in vivo* can be partially compensated by two related 2-oxoglutarate-dependent dioxygenases (2-OODs) (*Pelletier & Shirley, 1996*). One of the 2-ODD enzymes is FLS, catalyzing the double bond formation between C-2 and C-3 position of flavonols (*Pelletier & Shirley, 1996*; *Prescott et al., 2002*). Three Va-FLS isoforms were identified (Table S6), and showed distinct expression profiles in developing *V. amygdalina* leaves (Fig. 4). These expression results, coupled with either lack or low content in kaempferol and quercetin (Table S2), suggests that Va-FLS may have other functions in flavonoid biosynthesis. This view also was supported by the detailed mechanistic study in *A. thaliana* (*Turnbull et al., 2004*). Intriguingly, due to lack of intermediate metabolite of dihydrokaempferol in *V. amygdalina* leaves, how kaempferol is formed excited our interest. We observed a tight connection of molecular structure between kaempferol and apigenin (Fig. 4), and their contents showed a coordinated temporal pattern in developing *V. amygdalina* leaves (Table S6 ). Thus, it is hypothesized whether flavone 3-hydroxylase exist in *V. amygdalina* leaves. These data provide leads for future research on the interconnection of pathways involved in flavones and flavonols.

## CONCLUSIONS

In conclusion, metabolome analysis indicated abundant and diverse flavonoids in *V. amygdalina* leaves, including six dihydroflavones, 14 flavones, eight isoflavones, nine flavonols, two xanthones, one chalcone, one cyanidin, and 1 dihydroflavonol. Additionally, glycosylation and methylation commonly occur at -OH moieties of the C3 and C7 positions of flavonoid aglycones. By transcriptome sequencing, 60,422 genes were assembled, among which 31,872 genes were annotated at least in one of public databases. These sequencing results massively filled in the blanks in gene dataset of *V. amygdalina*. The temporal and comparative combination of metabolome and transcriptome results points to the key genes encoding regulatory enzymes involved in material supplying, flavonoid scaffold

biosynthesis, and flavonoid modifications. These findings in the present study would be conducive to understand the underlying mechanisms of flavonoid biosynthesis in *V. amygdalina*.

### Funding

This work was supported by the Innovative Research Team Program of Hainan Natural Science Fund (2018CXTD331) and Hainan University (KYQD(ZR)1701). The funders had no role in study design, data collection and analysis, decision to publish, or preparation of the manuscript.

### Grant Disclosures

The following grant information was disclosed by the authors:
Innovative Research Team Program of Hainan Natural Science Fund: 2018CXTD331.
Hainan University: KYQD(ZR)1701.

### Competing Interests

The authors declare there are no competing interests.

### Author Contributions

- Lanya Shui and Kaisen Huo performed the experiments, prepared figures and/or tables, and approved the final draft.
- Yan Chen performed the experiments, analyzed the data, authored or reviewed drafts of the paper, and approved the final draft.
- Zilin Zhang and Yanfang Li analyzed the data, authored or reviewed drafts of the paper, and approved the final draft.
- Jun Niu conceived and designed the experiments, prepared figures and/or tables, authored or reviewed drafts of the paper, and approved the final draft.

### Data Availability

  Data are available at NCBI BioProject PRJNA554198.

### Supplemental Information

Supplemental information for this article can be found online at http://dx.doi.org/10.7717/peerj.11239#supplemental-information.

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
