# Peer review of "Integrated metabolome and transcriptome revealed the flavonoid biosynthetic pathway in developing Vernonia amygdalina leaves"

_PeerJ, doi:10.7717/peerj.11239_

## Round 0.1 · original submission · Major Revisions

Dear authors,

As you can see, our reviewers identified several issues that need your attention. Please follow the recommendations of the reviewers carefully as they will check your revisions later.

Kind regards
Michael Wink
AE

Reviewer 1 ·

Basic reporting

Some English language corrections needed, especially in the Introduction See Minor Corrections:

P. 5 line 17 : what do you mean by "herb"? I think you mean "medicinal herb" instead

P. 5, l 31-34. Make a single phrase separated by comma.
By Illumina sequencing, the obtained over 200 million valid reads were assembled into 60,422 genes, among which 31,872 genes were annotated at least in one of public databases. Greatly increasing molecular resources devoted to V. amygdalina that lack genome sequence information.

P. 6, l.44 rapidly regenerating


Materials and Methods,
P. 7, l. 106. how were the time collection 10,20, 30 DAG points determined? Please explain in context of the plant’s life history.


P. 7, l. 105, Five plants located in

P. 7, l. 111, remove “The”, start the phrase “ 5g of …

The article is well structured.

The study is quite descriptive, the hypothesis that different flavonoids are present in different development stages is implied throughout the paper.

Caption Figure 6, correct please: "Proposed pathways of lavonoid" -> flavonoid

Experimental design

The data are original.

The question is relevant and fills a knowledge gap.

The investigation lacks some standards used in the RNAseq community. I point them out here:

Materials and Methods.

Why did the authors not use BUSCO for transcriptome completeness analysis?
This method is now standard, and is important to know the representation of the genes in the transcriptomes. It gives an idea how redundant their annotations are. A possible reference genome is sunflower.

Why do they use FPKM for transcript quantification, RSEM also gives out TPM, the latter is a better measure of expression for comparing samples. See : https://rna-seqblog.com/rpkm-fpkm-and-tpm-clearly-explained/

Why did they use actin as reference gene ? Please cite reference or else add the expression stability as a separate supplemental table/figure.

I always suggest to plot primer efficiency before calculating relative expression levels, as the base (number of amplification cycles needed) may not be 2 depending on primer binding efficiency. Specially for non-model species with unknown genomes, this base number cannot be just assumed and needs to be estimated for each primer.

P. 8, l.172. the Duncan method is a multiple rate test, therefore should be called : “Duncan MRT

P. 10, l. 223- 227. which statistical test was performed to determine DEG?
l. 227 : Venn diagrams are representations.
The analysis DE with multiple test correction: either Bonferroni, Benjamini-Hochberg, others see: https://nodepit.com/node/de.helmholtz_muenchen.ibis.ngs.edgeR.EdgeRNodeFactory

Validity of the findings

The other statistics are well performed.

Young leafs are better defended, therefore they show more flavonoid content and diversity.

I have suggestions for data representation and analyses to support their findings:

Figure 6 : I suggest showing the annotated unigenes that are of interest in panel b
only (yellow and red colors, i.e. p<0.05 and p< 0.01) and replacing unigenes by their annotated gene name. Move the large picture to supplemental files.
The same for panel C: replace unigenes by FMT-like , FGT-like etc. these names are more informative.

As a supplemental figure: it might be interesting to see alignments of the Unigenes identified in Table 1 (PAL), as multiple were mapped as the same Arabidopsis gene. Else are these assembly errors ?

Alignments might also allow a phylogenetic analysis for the multiple genes identified. The mapped genes to a sequenced Asteraceae species might be used for phylogenetic reconstruction, as for instance for Helianthus.

Else the presence of multiple copies for gene families that are usually single copy in other plants could be an indication of polyploidy.
Is there anything known about the plants’ ploidy level?

Additional comments

Dear author,

Your paper "Integrated metabolome and transcriptome revealed the flavonoid biosynthetic pathway in developing Vernonia amygdalina leaves"contains useful resources and information. Please consider the suggested corrections, for improvement.

Best regards!

Reviewer 2 ·

Basic reporting

The MS needs careful language revision so that statements are clear for their meaning and the draft is corrected for numerous grammatical errors.
More in-depth analysis is needed so that better biological inferences could be drawn, otherwise, the study is very superficial in spite of availability of all the raw data.

Experimental design

More information on statistical analysis is needed.

Validity of the findings

Statistical analysis is poorly explained. Details are missing in the text and in legends. Often it is not clear how many biological replicates were used, specially in RNA-Seq.

Additional comments

Review for the MS #55707 by Shui et al. submitted to PeerJ

The MS needs careful language revision so that statements are clear for their meaning and the draft is corrected for numerous grammatical errors.
The number of biological replicates used for RNA-seq and qRT-PCR needs to be specified clearly.

The authors claim that a change from 2.2% to 2.5% was a “significant” change in accumulation of flavonoids (line 181), but the statistics I Figure 1b does not support this claim. Moreover, the description of test statistic (which test was used, how many biological replicates were used, whether the P-value significance was post-hoc corrected; if so which tests, if not why) is completely missing.

I wonder the rational, why only 3 metabolites are specifically described (lines 184-188) and presented in Figure 2? This gives a misleading impression that the authors overlooked overall profiles. I find this description like putting the cart ahead of the horse. A better rational is required, or the description should be reorganised keeping in mind the next section.
In the abstract, the authors claim that they identified 42 flavonoids. In the results section, the authors describe the patterns of their accumulation in lines 189-210. This component forms an important part of the study of the authors. I wonder why the authors did not choose to present their results in the main body in forms of figures such as heatmaps, Venn diagrams etc, as they do for the genes in the next figures. Figure 2 should better represent the “dynamic” patterns of change of metabolites over time.
The overarching claim that “Our LC-MS analysis provides a metabolic basis for further exploration on the flavonoid biosynthetic pathway in V. amygdalina leaves.” should be supported by relevant figures and the data could be better analyzed and represented to justify this claim.
Moreover, as the authors introduce the importance of secondary metabolites of this plant, including its several medicinal usages, the authors should describe which unique metabolites they identified in their profiling study. The authors should also describe unique genes. The results of RNA-Seq mostly focuses on few very well known genes/ gene families. I presume, the motivation of such description was to show that they were able to identified well characterised genes. But then the study should be extended to new/novel intermediate metabolites and associated genes. This would only add value to the study.

Overall, I think that there is significant amount of data generated in this study. The data might be better represented and analysed to utilise its full power. The MS might further benefit from some reanalysis and reorganisation as well as careful English editing.

---

## Round 0.2 · Minor Revisions

Dear Authors,

Have a look at the comments of the reviewer and improve your manuscript accordingly.

Regards
Michael Wink
Academic Editor

Reviewer 1 ·

Basic reporting

English was largely improved.

Little corrections needed, as indicated in their submitted word document.

There's room for improvement in Figure 2. Comments below.

All else is well.

Experimental design

Improvements to previous version were made.

In the "Response to reviewers documents" they have some new information in the form of a table and two Figures, labeled Table A1, Figure A1, Figure A2. This is useful information, which should be included in the supplemental Figures.

All technical standards were met.

Validity of the findings

Ok.

Additional comments

Dear Author,

Your manuscript was greatly improved from the previous version.

The only suggestion for correction is the title and an improvement for Figure 2:

Title should read: « Heatmap of flavonoid profiles in Vernonia amygdalina leaves. The data were calculated as Log10(Peak area+0.01). »

Please add a hierarchical clustering below the heat map.

Please place names horizontal and use abbreviations if needed- as names are too long. Explanations of the abbreviations can be included as Figure foot.

Else, nice work!

Best wishes!

Annotated reviews are not available for download in order to protect the identity of reviewers who chose to remain anonymous.

---

## Round 0.3 · accepted · Accept

Dear authors
your revision has met the recommendations of the reviewers. The revision is adequate. Well done. Therefore, we can accept your contribution.

Kind regards,

Michael Wink
Academic Editor